# Components of effective letters of recommendation: A cross-sectional survey of academic faculty

Halah Ibrahim[1]*, Mohamad Kasem Mohamad[2], Shahad Abasaeed Elhag[2], Khairat Al-Habbal[1], Thana Harhara[2], Mustafa Shehadeh[1], Leen Oyoun Alsoud[1], Sawsan Abdel-Razig[3]

1 Department of Medical Science, Khalifa University College of Medicine and Health Sciences, Abu Dhabi, United Arab Emirates, 2 Department of Medicine, Sheikh Khalifa Medical City, Abu Dhabi, United Arab Emirates, 3 Department of Medicine, Cleveland Clinic Abu Dhabi, Abu Dhabi, United Arab Emirates

* halah.ibrahim@ku.ac.ae

**Data Availability Statement:** The datasets generated and/or analysed during the current study are available as S2 Appendix.

## Abstract

### Introduction

Conventional merit-based criteria, including standardized test scores and grade point averages, have become less available to residency programs to help distinguish applicants, making other components of the application, including letters of recommendation (LORs), important surrogate markers for performance. Despite their impact on applications, there is limited published data on LORs in the international setting.

### Methods

A cross-sectional survey of academic faculty was conducted between 9 January 2023 and 12 March 2023 at two large academic medical centers in the United Arab Emirates. Descriptive statistics were used to tabulate variable frequencies.

### Results

Of the 98 respondents, the majority were male (n = 67; 68.4%), Western-trained (n = 66; 67.3%), mid-career physicians (n = 46; 46.9%). Most respondents (n = 77; 78.6%) believed that the purpose of an LOR was to help an applicant match into their desired program. Letters rarely included important skills, such as leadership (n = 37; 37.8%), applicant involvement in research (n = 43; 43.9%), education (n = 38; 38.8%), or patient advocacy (n = 30; 30.6%). Most faculty (n = 81; 82.7%) were not familiar with standardized letters of recommendation. Only 7.3% (n = 7) of respondents previously received training in writing LORs, but 87.7% (n = 86) expressed an interest in this professional development opportunity.

### Conclusion

There is variability in perceptions and practices related to LOR writing in our international setting, with several areas for improvement. Given the increasing importance of LORs to a candidate's application, faculty development is necessary.

**Funding:** The author(s) received no specific funding for this work.

**Competing interests:** The authors have declared that no competing interests exist.

## Introduction

Traditional markers of academic merit, including grade point averages (GPAs) and standardized test scores, have become less discriminatory as many medical schools worldwide are adopting a pass/fail system during the pre-clinical years and examinations, such as the United States Medical Licensing Examination (USMLE Step 1), have transitioned to pass/fail [1, 2]. Other components of a candidate's application, including letters of recommendation (LORs), have, thereby, gained importance in match decisions, as applicants try to distinguish themselves from each other [3]. A large body of literature highlights the limitations of LORs, including poor inter-rater reliability, grade inflation, and even plagiarism by the letter writers [4–6]. Despite these concerns, Program Directors (PDs) consistently report using LORs in deciding whom to interview and rank [5]. Therefore, it is imperative that the faculty who write them are knowledgeable about their purpose and essential components.

LORs may be particularly consequential for the applications of international medical graduates (IMGs), who already face many challenges in the residency match [7]. As such, we sought to explore international faculty knowledge and experiences in LOR writing. We surveyed clinician-educators at two large academic medical centers to identify common practices, challenges, and areas for improvement in the recommendation letter writing process. By doing so, we hope to contribute to the ongoing discussion about the role and effectiveness of LORs in the residency application process from the perspective of an international medical education and healthcare system.

## Methodology

We conducted a cross-sectional survey of clinician-educators, between 9 January 2023 and 12 March 2023, at two hospitals. The Institutional Review Boards of both hospitals reviewed and approved this study. The *Strengthening the Reporting of Observational Studies in Epidemiology* (STROBE) checklist for cross-sectional surveys was used to guide our reporting [8].

### Setting and population

Two large academic medical centers in Abu Dhabi, United Arab Emirates were the sites for this study. They serve as primary teaching hospitals for medical students and are the country's largest sponsors of postgraduate training (internship, residency, and fellowship programs). In the UAE, most medical education programs follow competency-based training models [9, 10]. Medical school graduates can complete their training within the UAE or seek residency and fellowship training abroad. Study participants included all PDs, associate PDs, clerkship directors, and core teaching faculty. These individuals were selected to participate because of their roles in student and resident education and their experience in both reviewing and writing LORs.

### Survey instrument

The survey instrument was developed after a comprehensive review of the published literature on LORs and the residency application and match process; and then iteratively revised by a team of four investigators with over 50 combined years of experience as PDs and core faculty in both United States and international medical education. To optimize content validity, the research team included elements of recommendation letters considered important by PDs that were included in standardized letters developed for emergency medicine and internal medicine residency applications. Trainees on the research team provided input based on their recent experience with obtaining LORs. The questionnaire was piloted for length and

interpretability in September 2022 on 10 clinician-educators in a hospital that did not participate in the study. Textual changes were made based on their comments.

The final version consists of 25 questions, divided into 5 sections. To mitigate survey fatigue, most questions are in either a yes/no format or a 4-point Likert scale (never, sometimes, most of the time, always). Following basic demographic questions, participants were asked about perceptions of the value and purpose of LORs, prior experiences in writing letters, details of their current letter-writing practice (including information typically included), and knowledge and prior training in composing recommendation letters. (S1 Appendix).

### Data collection

After approval by each hospital's institutional review board, e-mail addresses of all PDs, associate PDs, and core faculty were obtained from the hospitals' education departments. In January 2023, each educator received an e-mail invitation and an individual link to an online survey. The e-mail described the purpose of the study and explained that it was anonymous and confidential. E-mail reminders were sent every 2 weeks, with a total of 3 reminders. Participation was voluntary, and no incentives were offered. Consent to participate in the study was indicated by the completion and return of the survey.

### Data analysis

Data were analyzed using Excel 2019. (S2 Appendix) Descriptive statistics to determine the frequencies of the various variables were used and tabulated as percent (%) and actual numbers (n). For Likert scale analysis, responses were dichotomized into always/most of the time and sometimes/never.

## Results

### Demographics of survey respondents

Of 129 surveys distributed, 98 complete responses were obtained (76% response rate). Table 1 lists participant demographics. The majority of respondents were male (n = 67; 68.4%), consistent with faculty gender ratios in the hospitals. Most respondents trained in Western countries; one third (n = 33; 33.7%) trained in the US. Nearly half (n = 46; 46.9%) of the respondents were mid-career physicians with 10–20 years of experience. The sample included a diverse representation of specialties, with the highest proportion of faculty (n = 36; 36.7%) coming from internal medicine, family medicine, and medical subspecialties, which represent the largest training programs in the hospitals (Table 1).

### Perceptions of LORs

The majority of respondents (n = 77; 78.6%) believed that LORs were important components of an application and that the primary purpose of an LOR was to help applicants match into their desired program (n = 65; 66.3%) (Table 2).

### Prior experiences in writing LORs

When asked about LOR writing practices (Table 3), only 20.4% (n = 20) of survey participants noted using a template for LORs. Approximately half (n = 52; 53.1%) of the respondents had declined requests for writing LORs; with the most common reasons cited being not knowing the applicant well (n = 40; 40.8%), lack of time (n = 30; 30.8%), and inability to provide a positive or helpful letter to the applicant (n = 25; 25.0%). While 36.7% (n = 36) of respondents admitted to using the same letter of recommendation for different people, only 2% (n = 2)

**Table 1. Demographics of survey respondents (N = 98).**

| Demographic | | n | %N |
|---|---|---|---|
| **Gender** | Male | 67 | 68.4% |
| | Female | 31 | 31.6% |
| **Training Country** | United States | 33 | 33.7% |
| | United Kingdom | 21 | 21.4% |
| | Canada | 12 | 12.2% |
| | UAE | 11 | 11.2% |
| | Asia | 8 | 8.2% |
| | Europe (not UK) | 8 | 8.2% |
| | Middle East/North Africa (Not UAE) | 5 | 5.1% |
| **Years of Practice** | 0 to 10 | 34 | 34.7% |
| | 10 to 20 | 46 | 46.9% |
| | 20 to 30 | 18 | 18.4% |
| **Specialty** | Family Medicine/Internal Medicine/Medical Subspecialty | 36 | 36.7% |
| | Anaesthesia/ Obstetrics and Gynecology/ Surgery/ Surgical Subspecialty | 32 | 32.7% |
| | Pediatrics/Pediatric subspecialty | 14 | 14.3% |
| | Other | 16 | 16.3% |
| **Current Role** | Core Teaching Faculty | 56 | 57.1% |
| | Associate Program Director or Program Director | 22 | 22.4% |
| | Chief of Division or Chair of Department | 12 | 12.2% |
| | Clerkship Director | 1 | 1.0% |
| | Other | 7 | 7.1% |
| **Average number of letters of recommendation written annually** | 1–5 | 75 | 76.5% |
| | 6–10 | 12 | 12.2% |
| | >10 | 6 | 6.1% |
| | None | 5 | 5.1% |

reported that they had copied letters of recommendation and 8.2% (n = 8) have asked an applicant to write their own LOR (Table 3).

## Current practices in writing LORs

While most faculty (n = 89; 90.8%) describe the nature of their relationship with the applicants, only 77.6% (n = 76) of respondents indicate the duration of the relationship. In their LORs, faculty most frequently comment on applicants' communication skills (n = 90; 91.9%), medical

**Table 2. Faculty perceptions of letters of recommendation (N = 98).**

| | | n | %N |
|---|---|---|---|
| **What do you think is the purpose of a letter of recommendation?** | To help applicants match into their desired program | 65 | 66.3% |
| | To provide an accurate assessment of the applicant | 27 | 27.6% |
| | A not-so-important part of the residency application process | 2 | 2.0% |
| | I don't know | 1 | 1.0% |
| | Other | 3 | 3.1% |
| **In your opinion, how important is a letter of recommendation for an applicant's chances of matching into a residency or fellowship program?** | Important | 77 | 78.6% |
| | Not Important | 21 | 21.4% |

**Table 3. LOR writing practices (N = 98).**

| | Yes | | No | |
|---|---|---|---|---|
| | **n** | **%N** | **n** | **%N** |
| **Have you said no to a letter of recommendation request?** | 52 | 53.1% | 46 | 46.9% |
| **Have you ever used the same letters of recommendation for different people?** | 36 | 36.7% | 62 | 63.3% |
| **Do you have a template for letters of recommendation?** | 20 | 20.4% | 78 | 79.6% |
| **Have you ever asked an applicant to write their own letter of recommendation?** | 8 | 8.2% | 90 | 91.8% |
| **Have you ever copied a letter of recommendation?** | 2 | 2.0% | 96 | 98.0% |

knowledge (n = 89; 90.8%), attitude (n = 86; 87.6%), and patient management skills (n = 85; 86.7%). The least represented components of LORs were areas for improvement (n = 12; 12.2%), extracurricular activities (n = 28; 28.6%), and patient advocacy (n = 30; 30.6%) (Table 4).

## Faculty knowledge and training in writing LORs

The vast majority of respondents were not familiar with standardized formats for LORs (n = 81; 82.7%) and were unaware of which programs require them (n = 88; 89.8%). Accordingly, the majority of faculty have never used a standardized template to write a LOR (n = 87; 88.8%).

Finally, only 7.1% (n = 7) of respondents have received prior training in writing LORs, with a majority (n = 86; 87.7%) expressing an interest in this training opportunity, most preferring webinar/online formats (n = 64; 65.3%) over in-person lectures/courses (n = 18; 18.4%).

**Table 4. Elements included in letters of recommendation (N = 98).**

**Do you include/comment on the following in your letters of recommendation?**

| | Always/Most of Time | | Sometimes/Never | |
|---|---|---|---|---|
| | **n** | **%N** | **n** | **%N** |
| Communication skills | 90 | 91.9% | 8 | 8.2% |
| Nature of Relationship | 89 | 90.8% | 10 | 10.2% |
| Medical knowledge | 88 | 89.8% | 10 | 10.2% |
| Behavioural/attitudinal | 86 | 87.8% | 12 | 12.2% |
| Problem solving and patient management | 85 | 86.7% | 13 | 13.3% |
| Duration of relationship with applicant | 76 | 77.6% | 22 | 22.4% |
| Procedural skills | 71 | 72.4% | 27 | 27.6% |
| Global assessment | 67 | 68.4% | 31 | 31.6% |
| Qualifications for the position | 67 | 68.4% | 31 | 31.6% |
| Work ethic | 60 | 61.2% | 38 | 38.8% |
| Teamwork | 55 | 56.1% | 43 | 43.9% |
| Sense of responsibility | 50 | 51.0% | 48 | 49.0% |
| Motivation | 48 | 49.0% | 50 | 51.0% |
| Specific examples of abilities | 47 | 48.0% | 51 | 52.0% |
| Intellectual curiosity | 44 | 44.9% | 54 | 55.1% |
| Research | 43 | 43.9% | 55 | 56.1% |
| Involvement in education | 38 | 38.8% | 61 | 61.2% |
| Leadership | 37 | 37.8% | 61 | 62.2% |
| Patient advocacy | 30 | 30.6% | 68 | 69.4% |
| Extracurricular activities | 28 | 28.6% | 70 | 71.4% |
| Areas for improvement | 12 | 12.2% | 86 | 87.8% |

## Discussion

LORs are an integral part of applicant selection for residency and fellowship programs. They should serve as an essential resource for information not available elsewhere in the application. The transition to pass/fail in many undergraduate courses and standardized tests has resulted in fewer objective assessments available to selection committees, placing greater emphasis on LORs. Accordingly, PDs have recently expressed concern that IMGs will be further disadvantaged in the residency and fellowship application process [3, 11]. There is a paucity of data from the international sites from which the IMGs apply or from the letter writers' perspectives. Our survey of educators in two large academic health centers identified numerous gaps in the letter-writing practices of international faculty that may adversely impact a candidate's application.

Consistent with the literature, our study found that most letter writers have received little or no guidance on writing LORs that accurately assess a candidate's fit for a training program [12, 13]. Although the vast majority of the faculty surveyed recognized the importance of LORs, 66.3% of respondents believed that their purpose is to help candidates match into their desired program, with only a quarter of faculty responding that LORs should provide an accurate assessment of the applicant. This perception likely leads to grade inflation- a major challenge in LORs. One study found that in a sample of LORs for an emergency medicine residency program, over 95% of applicants were ranked in the top third of their class [14]. While including areas for improvement may mitigate some of this bias, only 12% of faculty respondents in our study included this element in their LORs. Professional development on LORs should, therefore, begin with an aligned understanding of their purpose and role.

The content of LORs can also be problematic. Most respondents report on the applicants' medical knowledge, patient care, and procedural skills, reiterating information that can readily be found elsewhere in the application. Several studies have highlighted the importance of references to work ethic, teamwork, and motivation in LORs [15–17], but only about half of our faculty consistently include this information in their letters. Also, only a minority of letter writers discuss the applicant's leadership, patient advocacy, or involvement in education or extracurricular activities. These are all missed opportunities to highlight a candidate's non-cognitive attributes that may predict future success in residency and beyond.

It is notable that most of the study participants were unfamiliar with standardized formats for reference letters or the programs that have adopted them. Several specialties have transitioned to the use of standardized letters as they are believed to improve inter-rater reliability and decrease bias [17]. A survey of 150 emergency medicine PDs found that standardized letters were the most important determinant of candidate selection for interviews [18]. The fact that 88.8% of survey respondents have never written a standardized letter may create a disadvantage for the students. This is an important area for professional development. We are reassured that the vast majority of respondents deny copying LORs from other sources. Plagiarism has been reported in LOR authors, with one study finding at least one plagiarized letter in approximately 12% of applications [6]. Further, only a small percentage of our survey participants reported asking the applicants themselves to write the LORs. This has been suggested as a contributing factor to the plagiarism noted in LORs [6].

The results of this study underscore the need for professional development in letter writing. Only 7.1% of survey respondents reported receiving prior training. Fortunately, 87.7% are interested in guidance on LOR writing. Based on our findings, we are planning a series of faculty development workshops on this topic. Here, we list several recommendations for letter writers that are consistent with best practices in the literature and also contain guidance pertinent to international letter writers. First, faculty must be selective in writing LORs, ensuring

they only accept this responsibility for trainees with whom they have had substantive and sufficient time periods to evaluate performance. Second, as specific hospitals, medical schools, and authors may not be recognizable to PDs and selection committees in other countries, letter writers should include a brief description of their institution, their role, and details about their relationship with the applicant. Information regarding program or institutional accreditation by an international organization, such as the Accreditation Council for Graduate Medical Education International or the Royal College of Physicians and Surgeons of Canada International, may help benchmark the quality of the applicant's educational experience. Also, both faculty and trainees must be aware of specialties that require standardized letters and ensure that these specific formats are followed. Further, letters should include personal attributes with specific examples. Moreover, studies show that specific words may signal particular interpretations by selection committees [17]. For example, a study of LORs for surgical residency training noted that an "excellent" candidate was not considered as highly as an "outstanding" applicant [17]. Letter writers need to be cautious and deliberate about wording, using clear and "uncoded" comparative language to avoid unintended consequences for the applicant. Finally, we recommend setting aside sufficient time to meet with applicants prior to writing LORs and to write personal and meaningful assessments. Cultural norms may promote the writing of overly flattering letters that contribute to grade inflation and have limited utility. Viewing the LOR as a fair and accurate evaluation and being transparent about this intent with the trainee will ultimately lead to higher quality LORs.

Most studies on LORs focus on US trainee and faculty experiences. Our study enriches the existing literature by examining perceptions and practices of letter writers in an international medical education system. Our findings should be viewed in light of some limitations. First, faculty in only two hospitals were surveyed; results may not be generalizable to other academic medical centers in the UAE or other countries. Also, the sample size of the study may not have been large enough to capture the full range of experiences and perspectives on LOR writing. Finally, we cannot rule out social desirability bias in the responses. The use of anonymous online surveys was intended to minimize this response bias.

## Conclusion

In this study of international clinician-educators, we found variability in perceptions and practices related to LOR writing and several areas for improvement. Given the increasing importance of LORs to a candidate's application, faculty development is necessary to help the letter writers provide honest, insightful assessments that acknowledge the unique attributes of each candidate.

## Supporting information

**S1 Appendix. Letters of recommendation survey.**
(PDF)

**S2 Appendix. Datasets.**
(PDF)

## Author Contributions

**Conceptualization:** Halah Ibrahim, Thana Harhara.

**Data curation:** Mohamad Kasem Mohamad, Shahad Abasaeed Elhag.

**Formal analysis:** Sawsan Abdel-Razig.

**Methodology:** Mustafa Shehadeh.

**Supervision:** Halah Ibrahim.

**Writing – original draft:** Halah Ibrahim, Mohamad Kasem Mohamad, Shahad Abasaeed Elhag.

**Writing – review & editing:** Khairat Al-Habbal, Thana Harhara, Mustafa Shehadeh, Leen Oyoun Alsoud, Sawsan Abdel-Razig.

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
