## [Decision Letter · Decision Letter 0]

5 Dec 2023

PONE-D-23-32302Components of Effective Letters of Recommendation: A Cross-Sectional Survey of Academic FacultyPLOS ONE

Dear Dr. Ibrahim,

Thank you for submitting your manuscript to PLOS ONE. After careful consideration, we feel that it has merit but does not fully meet PLOS ONE’s publication criteria as it currently stands. Therefore, we invite you to submit a revised version of the manuscript that addresses the points raised during the review process.

We look forward to receiving your revised manuscript.

Kind regards,

Amitav Banerjee, M.D.

Academic Editor

PLOS ONE

Journal Requirements:

4. We note that you have referenced Maruca-Sullivan PE, Lane CE, Moore EZ, Ross DA which has currently not yet been accepted for publication. Please remove this from your References and amend this to state in the body of your manuscript: Maruca-Sullivan PE, Lane CE, Moore EZ, Ross DA [Submitted]”) as detailed online in our guide for authors

Additional Editor Comments:

Please revise according to referee comments.

Reviewers' comments:

Reviewer's Responses to Questions

**Comments to the Author**

1. Is the manuscript technically sound, and do the data support the conclusions?

Reviewer #1: Yes

2. Has the statistical analysis been performed appropriately and rigorously? 

Reviewer #1: Yes

3. Have the authors made all data underlying the findings in their manuscript fully available?

Reviewer #1: Yes

4. Is the manuscript presented in an intelligible fashion and written in standard English?

Reviewer #1: Yes

5. Review Comments to the Author

Reviewer #1: Good effort to stimulate the topic which is not given importance and needs to be refined. The article will help to generate new ideas and may result in policy formulation. Other researchers can take this work forward.

6. PLOS authors have the option to publish the peer review history of their article (what does this mean?). If published, this will include your full peer review and any attached files.

Reviewer #1: **Yes: **Pramila Menon

---

## [Author Response · Author response to Decision Letter 0]

7 Dec 2023

Dear Dr. Amitav Banerjee, 

Thank you for consideration of our manuscript. We appreciate the academic editor and reviewer’s efforts. Please see our point-by-point responses to comments below.

Editor’s comment 1. When submitting your revision, we need you to address these additional requirements.

Please ensure that your manuscript meets PLOS ONE's style requirements, including those for file naming. The PLOS ONE style templates can be found at https://journals.plos.org/plosone/s/file?id=wjVg/PLOSOne_formatting_sample_main_body.pdf and https://journals.plos.org/plosone/s/file?id=ba62/PLOSOne_formatting_sample_title_authors_affiliations.pdf

Response: The revised manuscript meets PLOS ONE style.

Editor’s comment 2. Did you know that depositing data in a repository is associated with up to a 25% citation advantage (https://doi.org/10.1371/journal.pone.0230416)? If you’ve not already done so, consider depositing your raw data in a repository to ensure your work is read, appreciated and cited by the largest possible audience. You’ll also earn an Accessible Data icon on your published paper if you deposit your data in any participating repository (https://plos.org/open-science/open-data/#accessible-data).

Response: Noted

Editor’s comment 3. We note that you have indicated that data from this study are available upon request. PLOS only allows data to be available upon request if there are legal or ethical restrictions on sharing data publicly. For more information on unacceptable data access restrictions, please see http://journals.plos.org/plosone/s/data-availability#loc-unacceptable-data-access-restrictions. 

Response: The data set is now included with the manuscript as supplemental material.

a) If there are ethical or legal restrictions on sharing a de-identified dataset, please explain them in detail (e.g., data contain potentially sensitive information, data are owned by a third-party organization, etc.) and who has imposed them (e.g., an ethics committee). Please also provide contact information for a data access committee, ethics committee, or other institutional body to which data requests may be sent.

Response: There are no legal or ethical restrictions. The datasets are included as a supplement 2 Appendix . The cover letter now highlights this point.

Editors comment 4. We note that you have referenced Maruca-Sullivan PE, Lane CE, Moore EZ, Ross DA which has currently not yet been accepted for publication. Please remove this from your References and amend this to state in the body of your manuscript: Maruca-Sullivan PE, Lane CE, Moore EZ, Ross DA [Submitted]”) as detailed online in our guide for authors

Response: Please note that this reference has been published in Medical Education Journal in 2018. The reference is correct as written.

Editors comment 5. Please review your reference list to ensure that it is complete and correct. If you have cited papers that have been retracted, please include the rationale for doing so in the manuscript text, or remove these references and replace them with relevant current references. Any changes to the reference list should be mentioned in the rebuttal letter that accompanies your revised manuscript. If you need to cite a retracted article, indicate the article’s retracted status in the References list and also include a citation and full reference for the retraction notice.

Response: The reference list has been revised and I confirm that it is complete and correct in the revised manuscript version.

---

## [Editor Report · Decision Letter 1]

18 Dec 2023

Components of Effective Letters of Recommendation: A Cross-Sectional Survey of Academic Faculty

PONE-D-23-32302R1

Dear Dr. Ibrahim,

We’re pleased to inform you that your manuscript has been judged scientifically suitable for publication and will be formally accepted for publication once it meets all outstanding technical requirements.

Kind regards,

Amitav Banerjee, M.D.

Academic Editor

PLOS ONE

Additional Editor Comments (optional):

Revision is satisfactory.
---

## [Editor Report · Acceptance letter]

14 Jan 2024

PONE-D-23-32302R1 

PLOS ONE

Dear Dr. Ibrahim, 

I'm pleased to inform you that your manuscript has been deemed suitable for publication in PLOS ONE. Congratulations! Your manuscript is now being handed over to our production team.

Kind regards, 

on behalf of

Dr. Amitav Banerjee 

Academic Editor

PLOS ONE